# *Calomys callosus*: An Experimental Animal Model Applied to Parasitic Diseases Investigations of Public Health Concern

**DOI:** 10.3390/pathogens11030369

**Published:** 2022-03-17

**Authors:** Rafael Borges Rosa, Mylla Spirandelli da Costa, Samuel Cota Teixeira, Emilene Ferreira de Castro, Willyenne Marília Dantas, Eloisa Amália Vieira Ferro, Murilo Vieira da Silva

**Affiliations:** 1Rodents Animal Facilities Complex, Federal University of Uberlandia, Uberlandia 38400-902, Brazil; rafael.brosa@ufu.br (R.B.R.); myllaspirandelli@ufu.br (M.S.d.C.); emilene.castro@ufu.br (E.F.d.C.); 2Institute of Biomedical Sciences, Federal University of Uberlandia, Uberlandia 38405-318, Brazil; samuctx@gmail.com (S.C.T.); eloisa.ferro@ufu.br (E.A.V.F.); 3Institute Aggeu Magalhães, Fiocruz Pernambuco, Recife 50740-465, Brazil; dantaswillyenne@gmail.com

**Keywords:** *Calomys callosus*, animal model, parasitic diseases, public health, *Schistosoma mansoni*, *Toxoplasma gondii*, *Leishmania*, *Trypanosoma*

## Abstract

The appearance and spread of parasitic diseases around the world aroused the interest of the scientific community to discover new animal models for improving the quality and specificity of surveys. *Calomys callosus* is a rodent native to South America, an easy handling model, with satisfactory longevity and reproducibility. *C. callosus* is susceptible to toxoplasmosis and can be used as experimental model for the study the pathogenesis, treatment, vertical transmission, and ocular toxoplasmosis. *C. callosus* can also be used to study cutaneous and visceral leishmaniasis, as the animals present cutaneous lesions, as well as parasites in the organs. *C. callosus* has epidemiological importance in Chagas disease, and since it is a *Trypanosoma cruzi* natural host in which rodents show high parasitemia and lethality, they are also effective as a model of congenital transmission. In the study of schistosomiasis, *Schistosoma mansoni* was proven to be a *C. callosus* natural host; thus, this rodent is a great model for fibrosis, hepatic granulomatous reaction, and celloma associated with lymphomyeloid tissue (CALT) during *S. mansoni* infection. In this review, we summarize the leading studies of parasitic diseases that used *C. callosus* as a rodent experimental model, describing the main uses and characteristics that led them to be considered an effective model.

## 1. Introduction

*Calomys callosus* is a rodent species that belongs to the Cricetidae family, is native to South America, and is similar in size to mice [1,2,3,4]. The introduction of *C. callosus* as a laboratory animal occurred in 1967 by Petter et al. [5]. Since they were bred for this purpose, these rodents have characteristics such as easy handling, satisfactory longevity, and reproducibility [6,7,8]. Currently the biology, behavior, and anatomy of *C. callosus* are well known, as the species has been established in experimental laboratories and is considered an effective model to be used in infectious disease studies [5,9,10].

Studies with *C. callosus* showed that these animals are considered an adequate experimental model for investigating the dynamics of the acute phase and the acquired, ocular, and congenital forms of toxoplasmosis. Research has also shown that *C. callosus* is a tolerant natural host to *Trypanosoma*
*cruzi* infection, in addition to being considered an alternative model in leishmaniasis studies since it can be naturally infected by *Schistosoma mansoni* [11,12,13,14,15,16,17,18].

Despite all these works involving the use of *C. callosus* as a great model for the study infectious diseases of public health concern, this rodent is still underused. In this review, we summarize the main studies of parasitic diseases (Appendix A) that used *C. callosus* as a rodent experimental model and discuss the principal uses and characteristics that led these animals to be considered an effective model.

Our review has as its main objective the presentation of the applicability of the experimental model *C. callosus* in the study of different parasitic diseases of public health importance—diseases that burden health systems and cause significant mortality. Therefore, aiming to illustrate this variability in the application of the model, we used four parasites with marked pathophysiological differences, being one Apicomplexa, two Trypanosomatids, and one Helminth).

## 2. Toxoplasmosis

*Toxoplasma gondii*, an obligate intracellular protozoan parasite, is the etiologic agent of toxoplasmosis, one of the most common zoonotic food-borne infection that affects up to one-third of the global human population [19]. Acquired infection with *T. gondii* is usually asymptomatic in healthy individuals; however, the primary infection or reactivation of chronic infection, caused by this opportunistic human pathogen, can cause severe clinical features in immunocompromised individuals, human fetuses, and newborns [20,21].

*T. gondii* has a notable ability to infect efficiently target organs, including brain, eyes, and placenta [22], which configures a limiting factor for obtaining infected human material. Thus, it is mandatory to use experimental animal models capable of mimicking the course of toxoplasmosis in order to elucidate several aspects of the mechanisms responsible for pathological lesions associated with the clinical disease.

In this context, a considerable number of studies have shown the rodent *C. callosus* to be an efficient experimental model for investigating the underlying mechanisms associated with *T. gondii* infection. For this reason, we consider it appropriate to compile these studies addressing a current view, for the first time, of the remarkable findings involving the use of *C. callosus* as an experimental model, especially for understanding the acute phase and the acquired, ocular, and congenital forms of the toxoplasmosis.

### 2.1. Acute Phase of Toxoplasmosis

The first study addressing the use of the *C. callosus* as a promising model to investigate the experimental infection by *T. gondii* was published in 1998 by Favoreto-Junior et al. In this work, *C. callosus* intraperitoneally infected by *T. gondii* tachyzoites of RH strain were demonstrated to be highly susceptible to parasitic infection, which was characterized by a massive presence of extracellular parasites in the peritoneal cavity and by a high parasite load in spleen, lung, intestine, brain, and kidney [23]. In a later study, *C. callosus* was demonstrated to be an interesting tool for investigating the role of mast cells in the acute phase of inflammatory response caused by *T. gondii* [24]. Thus, these early works showing *C. callosus* as a host with susceptibility to *T. gondii* infection initiated new ways of investigating in depth the parasitic acute infection pathogenesis, as well as the host immune response during toxoplasmosis. In this scenario, Franco et al. (2014) studied the different rate of susceptibility between *C. callosus* females and males during acute infection of Brazilian atypical *T. gondii* strains. These authors demonstrated that this rodent is susceptible to parasite infection with gender differences in severity of infection, regardless of *T. gondii* strain [25].

### 2.2. Acquired and Congenital Ocular Toxoplasmosis

In humans, *T. gondii* is an important cause of ocular disease in immunosuppressed and immunocompetent individuals, especially in newborns [26]. The ocular toxoplasmosis is one of the most common clinical manifestations caused by *T. gondii* infection, which results in retinochoroiditis in 75–80% of cases, while acquired systemic infection can cause ocular disease in 1% to 3% of individuals, specifically the unilateral form [27,28]. However, although ocular lesions in patients with acquired toxoplasmosis appear to be uncommon, several studies have reported a high incidence of ocular disease during toxoplasmosis outbreaks [29,30]. Despite the importance, the knowledge about the complex pathogenesis of ocular toxoplasmosis is still unclear [31], thus, highlighting the urgency to find new study models for understanding this disease.

In this scenario, Pereira et al. (1999) demonstrated that males and virgin females (acquired form) and pregnant females (congenital form) of *C. callosus* orally infected with *T. gondii* cysts (ME49 strain) developed severe ocular lesions, which were associated with the presence of cysts, free tachyzoites, and inflammatory cells in the retina. Interestingly, as shown in humans, the authors reported that most animals with acquired systemic toxoplasmosis had unilateral ocular cysts post-infection [32].

About the pathogenesis of ocular toxoplasmosis, it was shown that *C. callosus* enucleated eyes presented a high number of mast cells after intraperitoneal or conjunctival inoculation with *T. gondii* tachyzoites (RH strain) [33]. Surprisingly, some studies have implicated the presence of ocular mast cells and their connection to hypersensitivity reactions with development of human ocular toxoplasmosis [31,34]. Thus, due to both the difficulty in obtaining human eyes removed from patients with ocular toxoplasmosis and the great similarity in comparison with human samples, the rodent *C. callosus* has emerged as a remarkable experimental model for studying mechanisms associated with ocular diseases, especially ocular toxoplasmosis.

### 2.3. Congenital Toxoplasmosis

The search for experimental models for assessing congenital toxoplasmosis have been explored since the 1950s [35,36]. In humans, the vertical transmission of *T. gondii* from the mother to the fetus can result in fetal death, abortion, or severe clinical manifestation in infected fetuses and newborns, including neurologic damage, neurocognitive deficits, or chorioretinitis, among others [21,37]. In this context, a pioneering study by Ferro et al. (1999), using *C. callosus* as experimental model, demonstrated that *T. gondii* is capable of infecting trophoblast cells in the early stage of pregnancy, highlighting the trophoblast role as a possible route of access from maternal to embryonic tissues [38].

It was reported that *T. gondii* vertical transmission occurs in acutely infected pregnant *C. callosus*, but not in the infection’s chronic phase [11]. Furthermore, as shown in pregnant women, the susceptibility to *T. gondii* vertical transmission is temporally dependent on preconceptional infection in *C. callosus*—i.e., parasitic infection acquired during or just before pregnancy can result in vertical transmission in this model [39]. For years, several studies showed that primary *T. gondii* infection is capable of leading a life-long immunity that can prevent vertical transmission [40,41]. However, some studies have reported cases of congenital toxoplasmosis in fetuses to immunocompetent mothers who had been infected with the parasite before conception [42,43,44,45]. In line with this discussion, Franco et al. (2015) showed that *C. callosus* females chronically infected with *T. gondii* when reinfected during pregnancy resulted in vertical transmission [46].

Despite the existence of some drugs for treating human congenital toxoplasmosis, it is widely known that these compounds are associated with serious side effects to both mothers and fetuses [20,47]. Moreover, the treatment options available for congenital toxoplasmosis suppress the active infection but do not cure the latent infection [48,49]. Thus, the conventional therapy is still limited, which affects mortality and quality of life on pregnancy and newborn health [50]. For this reason, the search for new bioactive substances for congenital toxoplasmosis treatment gathered great interest in the past few decades [51,52,53,54]. In this scenario, some studies have suggested the use of *C. callosus* as an interesting experimental model for assessing the efficacy of new possible compounds that have remarkable activities for controlling *T. gondii* infection, as well as for inhibiting vertical transmission [55,56,57].

## 3. Leishmaniasis

Leishmaniasis is an infectious disease caused by a protozoa parasite, *Leishmania* sp., that affects economically vulnerable populations and represents a significant problem in public health because it is the second most neglected tropical disease in mortality and seventh in disability [58]. *Leishmania* parasites are transmitted through the bites of infected female phlebotomine sandflies to more than 70 animal species, including humans [59].

Highly endemic regions such as Algeria, Brazil, Iraq, Somalia, and Sudan have more than 1 billion people who are at risk of infection. It is estimated that there are 700,000–1.2 million new cases and 14,000–40,000 deaths from the disease each year worldwide. Available data from 2018 show that out of the 200 countries and territories that reported to the World Health Organization, 97 (49%) were considered endemic for leishmaniasis [60]. In other words, it is a widespread concern around the world.

Leishmaniasis presents itself in three clinical forms: visceral leishmaniasis (VL)—affects the internal organs and causing more systemic symptoms such as fever, swelling of liver and spleen, weight loss, and petechiae; cutaneous leishmaniasis (CL)—is the most common form and affects the skin, causing a small lump or sores to appear at the bite site; and mucocutaneous leishmaniasis—leads to partial or total destruction of mucous membranes of the nose, mouth, and throat [61].

### 3.1. Visceral Leishmaniasis (VL)

Also known as kala-azar, this leishmaniasis clinical form is fatal if not treated in over 95% of cases, and it presents irregular bouts of fever, weight loss, spleen and liver enlargement, and anemia. The locations with higher incidence are Brazil, East Africa, and India. An estimated 50,000 to 90,000 new cases of VL occur worldwide annually, but only 25% to 45% are reported to WHO. It remains one of the neglected parasitic diseases, with outbreak and mortality potential. In 2019, more than 90% of new cases reported to WHO occurred in 10 countries: Brazil, Ethiopia, Eritrea, India, Iraq, Kenya, Nepal, Somalia, South Sudan, and Sudan [60].

Experiments performed using *Leishmania infantum*, the etiological agent of VL, demonstrated that after 3 months of infection *C. callosus* presented hepato and splenomegaly, also parasite amastigote forms were found intra (macrophages) and extracellularly in the bone marrow, liver and spleen smears of infected rodents [17]. *C. callosus* could also be useful as an experimental animal model for VL, due to histopathological and visceral alterations promoted after infection.

### 3.2. Cutaneous Leishmaniasis (CL)

CL is the most common leishmaniasis form and causes skin lesions, mainly ulcers on exposed body parts, leaving life-long scars and serious disability or stigma. About 95% of CL cases occur in the Americas, the Mediterranean basin, the Middle East, and Central Asia. In 2019, over 87% of new CL cases occurred in 10 countries: Afghanistan, Algeria, Brazil, Colombia, Iran (Islamic Republic of), Iraq, Libya, Pakistan, the Syrian Arab Republic, and Tunisia. It is estimated that between 600,000 to 1 million new cases occur worldwide annually [60].

Mello (1978) investigated the infection of some parasitic strains, including *Leishmania amazonensis*, isolated from the wound of a patient affected by CL. *C. callosus* presented lesions at the application sites from the 40th day post-infection, and the parasite presence was confirmed by lesion smear [16]. This rodent has the ability to reproduce CL lesions, proving to be a decent model for the study of this leishmaniasis clinical form.

Despite the fact that mice and hamsters are still the most commonly used rodents as experimental animal models in leishmaniosis studies [62,63,64,65] and that there are few studies using *C. callosus*, here we demonstrate that *C. callosus* should be potentially used as an experimental model in leishmaniasis research [16,17].

## 4. Chagas Disease

Chagas disease, also known as American trypanosomiasis, is a potentially fatal disease caused by the protozoan parasite *Trypanosoma cruzi* [66]. It is estimated that 6 million people worldwide are infected, mainly in the endemic areas of 21 countries in Latin America. Mortality associated with the disease is higher than other parasitic diseases, causing approximately 10,000 deaths per year, in addition to more than 75 million people who are exposed to the risk of contracting the infection [67].

*T. cruzi* was discovered and described for the first time by the Brazilian physician and scientist Carlos Chagas, who investigated the hematophagous *Panstrongylus megistus*. Monkeys of the species *Callithrix penicillata* were exposed to the bite of vectors collected in the field, and it was found through hematological analysis that the animals were infected with the parasite of the genus *Trypanosoma*, whose life cycle was later elucidated and then identified as the species *T. cruzi*, now known as the etiologic agent of Chagas disease [66,68].

The main forms of Chagas disease transmission are contact with infected triatomines feces after being bitten and oral ingestion of contaminated food with parasites from infected triatomines (açai, sugar cane, etc.); congenital transmission occurs by the parasites passage from women infected with *T. cruzi* to their infants during pregnancy or childbirth, by blood transfusion or organ transplantation from infected donors to healthy recipients, and by accidental contact by injured skin or mucous membranes with contaminated material during handling in the laboratory [69].

Chagas disease has two distinct phases: acute and chronic. The prognosis and symptoms for these phases are different [70]. The acute phase is characterized by trypomastigotes presence circulating in the peripheral blood, prolonged fever, headache, severe weakness, and swelling of the face and legs; physiological changes are edema, lymphadenopathy, anemia, hepatosplenomegaly, abnormalities in the electrocardiogram (ECG), involvement of the central nervous system, and death [71]. The chronic phase usually starts with an asymptomatic period; if progressing, individuals may have cardiac involvement, fever, cardiomegaly, apical aneurysms, and/or ECG abnormalities. In addition, fever, megaesophagus, megacolon due to myenteric plexus destruction, and constipation [72].

For the last 50 years, the drugs of choice have been benznidazole and nifurtimox for *T. cruzi* infection treatment. However, they have a long period of treatment and a cure rate around 70%, in addition to high liver and kidney toxicity and being contraindicated for pregnant women. Adverse reactions such as allergic dermatitis, peripheral neuropathy, anorexia, weight loss, and insomnia are some of the main reasons why many patients drop out of treatment. The search for new drug treatments involves more effective and less toxic drugs to replace the usual and also studies to create a vaccine have been developed [68].

*C. callosus* has epidemiological importance in Chagas disease since it is a *T. cruzi* natural host and can be considered a good experimental laboratory model due to its high permissiveness to *T. cruzi* infection and its ability to maintain a patient, regular, and long-term parasitemia [73,74,75]. Although *Rattus rattus*, *Mus musculus,* and the opossums of the genus *Didelphis* spp. can become infected with *T. cruzi,* as demonstrated in several studies, they do not always represent the most important mammalian species for the maintenance of the parasite [76].

Historically, the natural occurrence of *T. cruzi* in *Calomys* rodent was first described in 1976 by Basso et al., who reported the parasite in *Calomys musculinus* captured in a rural area of the province of Córdoba, Argentina [77]. In Brazil, Mello and Teixeira (1977) reported the presence of *T. cruzi* in wild *Calomys expulsus* captured in Formosa, state of Goiás [78], and also identified a flagellated protozoan belonging to the sub-genus *Herpetosoma* in *C. callosus* wild. After evaluating other studies, the authors considered the parasite found in *C. callosus* to be the species *Trypanosoma lewisi* [79].

The collected strain of *T. lewisi* was experimentally inoculated into *C. callosus* under laboratory conditions, and subsequently, the blood of the animals was examined on alternate days. The evaluations showed that the animals were positive for the infection. It was also found that the rodents had high blood parasitemia that disappeared between 45 and 60 days after inoculation, demonstrating the ability to infect *C. callosus* [79].

Although Mello (1978) believed that there was a relationship of parasite–host specificity, it is currently known that T. Lewisi, although it has a higher rate of infection for Rattus rattus, is also capable of infecting several other wild rodent species [80]. The same has also been observed for primates [81], including man [82]. Therefore, it can be said that there is no parasite–host specificity.

### 4.1. Parasitemia and Lethality

In order to study *C. callosus* susceptibility to *T. cruzi* infection and characterize the prepatency period, rodents were infected with four different strains: R52, R64, R65, and M226. Daily blood tests were performed to detect trypomastigotes and follow the parasitemia evolution. R52, R64, and R65 strains present high parasitemia peaks and low lethality, while M226 strain showed low parasitemia peaks and lethality was zero [74]. Despite the different results between strains, the study showed positive results, concluding that the infective capacity was wielded in *C. callosus* regardless of strain and confirming this rodent as a good experimental model for Chagas disease [75].

### 4.2. Congenital Transmission

Mello and Borges (1981) also described the ability of *C. callosus* females to transmit the infection to their offspring. Female rodents in Chagas disease chronic phase started to reproduce, and all newborns were submitted to direct analysis of fresh blood and presented positive parasitemia. Although the transmission rate was low compared with intraperitoneal infection, in order to investigate *T. cruzi* in *C. callosus* different tissues, as in heart and leg muscle, infected animals have their organs collected between the 9th and 30th post infection day. Histopathology results presented lesions and parasite infiltration in cardiac and skeletal muscles [83]. Even with the low parasitism rate, it was possible to observe that there was *T. cruzi* congenital transmission and that these animals had infection sequelae, such as cardiac and muscle injuries. Thus, *C. callosus* can also be used as an animal model for congenital transmission for Chagas disease.

## 5. Schistosomiasis

Schistosomiasis is an infectious parasitic disease caused by trematode worms of *Schistosoma* genus. Considered a neglected disease, schistosomiasis transmission has been reported from 78 countries [84]. Studies on this parasite have been taking place for several decades, such as in Brazil, which since the 1950s has brought great contributions to the clinical and epidemiological aspects of the disease [85].

Schistosomiasis is a disease of tropical and subtropical areas, such as Africa, the Caribbean, Brazil, Venezuela, Suriname, China, Indonesia, Philippines, Cambodia, and Corsica, whose main form of transmission occurs with the penetration of the larva in the skin during contact with the infested water. After infection, the larvae develop into adult schistosomes that will live in the host’s blood [86]. There are five main species that cause schistosomiasis in humans—*S. mansoni*, *S. japonicum*, *S. mekongi*, *S. guineensis* (causing intestinal schistosomiasis), and *S. haematobium* (causing urogenital schistosomiasis)—and infection occurs in areas with low health coverage [87].

After larval skin penetration, the larvae develop into adult schistosomes, and adult worms live in the blood vessels and release eggs in the feces or urine to continue the lifecycle; transmission occurs when eggs, which hatch in water, contaminate freshwater sources [87].

Symptoms of schistosomiasis are caused by the reaction of the body to the *Schistosoma* eggs and include intestinal schistosomiasis, liver enlargement, accumulation of fluid in the peritoneal cavity, hypertension of the abdominal blood vessels, and spleen enlargement. Additionally, the condition of urogenital schistosomiasis features hematuria, fibrosis of the bladder and ureter, bladder cancer, and kidney damage and can lead to infertility. The chronic phase of schistosomiasis can result in death [88].

Due to the importance of this disease for world public health, different animal experimentation models have emerged to study the biology, immunology, and pathogenesis of schistosomiasis, such as rodents, lagomorphs, and primates of different species [89,90]. Rodents tend to be the animals of choice, essentially due to their easy availability, fecundity, and susceptibility to experimental infection [91]. An animal experimentation model discovered for investigating *Schistosoma* is *C. callosus*, first proven in an investigation of natural infection in 1979 with *S. mansoni* [18].

Swiss mice and the rodent *C. callosus* were used in a study to compare the prepatent period of Schistosoma infection. The animals were infected via the transcutaneous route, simulating a transdermal infection. The animals were exposed to the cercariae from human isolate for 45 min; the coprological method evaluation showed that the prepatent period observed in *C. callosus* was 43–44 days, while in Swiss mice it occurred 40 days after exposition [16]. The evaluation showed that *C. callosus* have a prepatent period equal to that of mice, which are experimental models known to be widely used in laboratories. Considering that the prepatent period occurs, on average, from thirty to sixty days after infection in human patients [92], the ability of *C. callosus* to present clinical signs similar to those observed in clinical cases of these patients is observed.

*C. callosus* was also shown to be a great fibrosis experimental model when infected with *S. mansoni*; between 55–160 days post infection, prominent intestinal subserous nodules with granular features were observed, and some had hemorrhage after their removal [93] and developed a severe hepatic granulomatous reaction, suggesting signs of hypersensitivity [94].

During schistosomiasis, *C. callosus* present an omental reactivity in milky spots with patches that represent subsidiary foci of lymphomyeloid tissue-associated coelom (CALT), which was also observed in mice [95,96]. *S. mansoni* worms live predominantly in mesenteric vessels, and consequently, their released products can reach the mesenteric and omental microcirculation, which leads to CALT [97].

A study using *C. callosus* was carried out to evaluate the effect of schistosomiasis in CALT during the acute, transient, and chronic phase of infection. The administration route was percutaneous, where *S. mansoni* cercariae were introduced. Omental milky spots were found in lymphomyelocytic (42–90 days post infection) and lymphoplasmacytic cells (160 days post infection); also, the milky spots are the preferred site of the germinal center, mast cell maturation, and proliferation in *C. callosus*, as observed in rats [98,99].

Administration routes that mimic natural infection and susceptibility to *S. mansoni* were demonstrated to be efficient through animal experimentation with *C. callosus*, confirming an important role of this rodent’s use in schistosomiasis research.

## 6. Challenges in Maintaining *C. callosus* as an Experimental Model

Considering the studies presented in this review, it is evident that *C. callosus* as an experimental model has great relevance for understanding the pathophysiology of several infectious diseases caused by parasites from different genera and species, thus contributing to the development of prophylactic and therapeutic measures. However, some factors such as legal issues associated with biosafety and the environment, animal welfare, breeding, transport, adaptation, and quarantine and maintenance in captivity can be obstacles, but not an impediment to the model’s use [100].

*C. callosus* is a wild rodent, and its capture and maintenance for scientific research purposes is directly related to the legal aspects of each country and must be strictly enforced. In Brazil, the competent authority is the Biodiversity Authorization and Information System (SISBIO), which can provide authorization for *C. callosus* capture. In addition, the experimental protocol must be evaluated and approved by the ethics committee [101].

After the approval of legally responsible authorities with ethical principles, species capture, maintenance, and breeding must be carried out in animal facilities. *C. callosus* must fulfill a quarantine period to ensure the health of the researchers and technicians who handle these animals. Thus, all biosafety protocols must be adopted, such as the use of collective and individual protection equipment, preventing the transmission of diseases and accidents [102].

Clinical evaluation and sanitary control must be performed by a veterinarian as a guarantee of animal health, as well as of the quality of the results obtained with their use in experimental tests. In the case of animals infected with zoonoses, they must be treated to eliminate the infection [103].

In addition to ethical issues related to animal welfare, for experimental results to be reproducible and reliable, an environmental enrichment program must be adopted, considering the physiological needs of the species to maintain homeostasis [104]. For this, we suggest that an ethogram be applied and that an intense literature review be carried out on the natural behaviors of the species prior to the implementation of the environmental enrichment program. Additionally, it is important that the animal facility has conditions to offer this species luminosity, temperature, humidity, and noise within the limits of comfort.

As for the other factors related to the maintenance and breeding of *C. callosus*, in a summarized way, a management approach similar to that used for other rodents such as rats and mice must be adopted. In daily handling, for example, avoid picking up animals by the tail—the use of tubes or cupped hands is highly recommended. When transporting, use boxes that guarantee safety and comfort to the animals [105,106].

To maintain the heterozygosity of the established colony and to avoid frequent captures of specimens in the wild, it is recommended to adopt mating systems used for outbreed colonies in animal facilities, such as random mate, Poiley method, and Falconer method [107,108]. These systems have their limitations, but they are widely accepted. Furthermore, it is possible to consider the adoption of inbreeding and establishment of an inbred lineage from the monitoring and characterization of allele frequency, which creates a sub-colony. In this case, it is necessary that this information be evidenced in the articles in order to contribute to the tests’ reproducibility.

## 7. Conclusions

Toxoplasmosis, Chagas disease, leishmaniasis, and schistosomiasis are parasitic diseases with worldwide impact. For a long time, the scientific community has been dedicated to study these parasitosis regarding their parasite biology, mechanism of infection, disease development, and novel treatments. For this, it is essential to use appropriate animal experimentation models to make the research accurate and specific. *C. callosus*, a species native from South America, is an animal model capable of efficiently replicating these parasitic diseases—a model with easy handling and with satisfactory longevity and reproducibility (Figure 1). Although, it is not widely used, these animals are considered effective models for carrying out any study in the parasitic diseases area involving these four main parasites.

## Figures and Tables

**Figure 1 pathogens-11-00369-f001:**
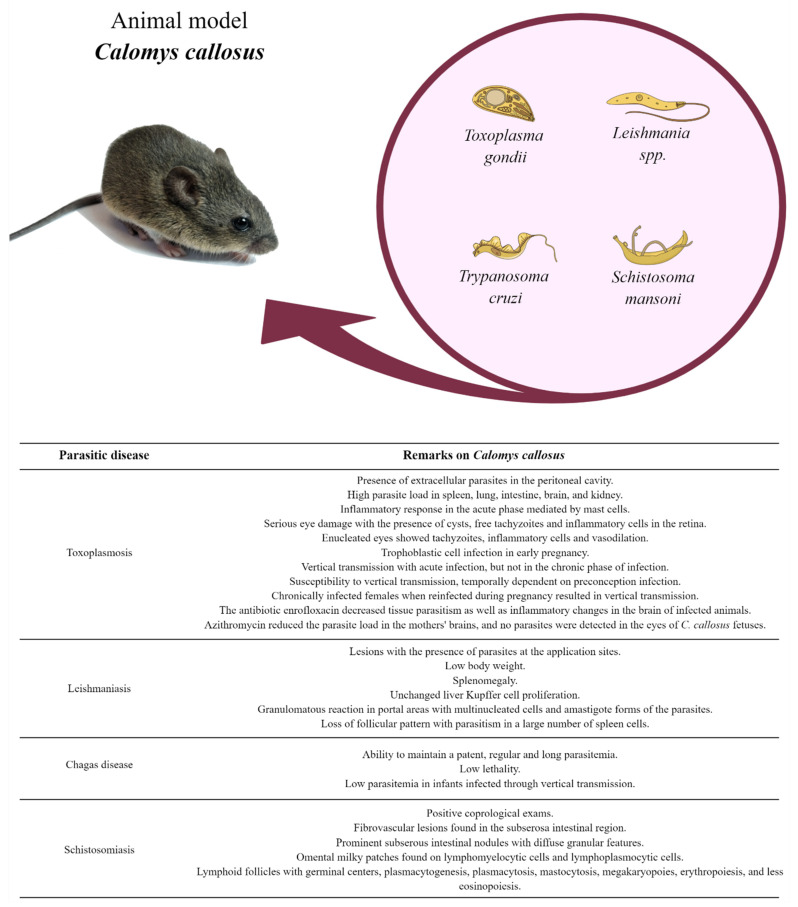
Remarkable points about *Calomys callosus* use as a model for animal experimentation. This figure lists the main experimental phenotypes observed in infected *C. callosus* according to different parasitic diseases such as toxoplasmosis, leishmaniasis, Chagas disease, and schistosomiasis. Created with MindtheGraph.com (accessed on 15 March 2022). All the pictures were taken by the authors.

## Data Availability

Not applicable.

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
