# Peer review of "Calomys callosus*: An Experimental Animal Model Applied to Parasitic Diseases Investigations of Public Health Concern"

_pathogens, 2022, doi:10.3390/pathogens11030369_

Round 1
Reviewer 1 Report
The authors review the studies that employed the wild rodent Callomys callosus as experimental model for different parasitic diseases. In this revision, authors present some biological aspects that encourage the maintenance of this rodent species in captivity and present the published results with experimental infection in distinct topics: toxoplasmosis, leishmaniasis, Chagas disease and Schistosomiasis. I have some points to be considered by the authors.
MAJOR POINTS
1) My main recommendation for authors is to include a topic of difficulties and challenges for the maintenance of a wild species in captivity. These challenges include the environmental and biosafety legal issues involved in the maintenance of a wild species in captivity, the field capture of the reproductive matrices (probably not in a single expedition), their transport, the adaptation, quarantine and maintenance in captivity, the necessary environmental enrichment, the care to avoid endogamy and the differences in the bioterism of this species compared to others conventionally used in scientific research, among others.
2) The definition of what constitute a reservoir host depends on many variables, such as geographical origin, interaction with other mammals, presence of vectors, distribution within forest strata, environmental characteristics, etc. So, defining a reservoir is much more than just finds it infected. And this aspect was not yet described in a definitive manner for T. cruzi infection in C. callosus. Thus, I suggest authors to exclude the word "reservoir" when describing the infection by T. cruzi, keeping only "natural host". Revise it throughout the manuscript, including lines 17/18 (abstract), 38 (introduction) and 226/227 (Chagas disease topic).
3) Leishmaniasis: The article that cites Leishmania donovani chagasi is from 1984 and this is no longer a valid name. Considering the nomenclature described in the article and the origin of the sample (Maranhão State, Brazil) we can state that the Leishmania species used in the experimental infections was Leishmania infantum. Also Leishmania mexicana amazonensis is no longer a valid name for the species. The valid name is Leishmania amazonensis. Correct them in the manuscript (this topic and Table 1).
4) Chagas Disease - Lines 200/201: Although T. infestans was the main vector of the disease for many decades in Brazil, the insect examined by Carlos Chagas was a Panstrongylus megistus. I suggest reviewing and citing the original article in this paragraph (Chagas, 1909). In the same sentence: “the life cycle” is of the parasite and not of the disease. Correct it.
5) Trypanosoma (Herbesoma) marie (lines 235-240): First the correct subgenus name in the description is Herpetosoma. Second, this supposed new species was described at a time when descriptions were mostly based on morphological characters of the parasites, without confirmatory biochemical or molecular tests. This particular case is probably an erroneous description of a species that is not valid, given that this is the only description and the only study to date (more than 40 years later) that cites this Trypanosoma species. Moreover, the characteristic of an abrupt drop in parasitemia is commonly described for all species within Herpetosoma subgenus and the species found by the authors is probably Trypanosoma lewisi. This cosmopolitan parasite was considered species-specific for a long time and for this reason, a new nomenclature was sought for each new host. Currently, it is known that T. lewisi is capable of infecting many other hosts besides Rattus rattus, including wild rodents and primates, including man. Therefore, the course of infection does not point to a parasite-host specificity (line 240). This critical analysis has to be presented by the authors in this review.
6) Table 1 provides a methodological detail that is not important for the review. I suggest present this table as a supplementary material. In addition, it is important to correct the taxonomic status of the parasites as described above and to revise scientific names in italics.
MINOR POINTS
1) Keywords: Put scientific names in italics. Trypanosoma, not Tripanosoma
2) Figure 1: Replace “Leishmania ssp” by “Leishmania sp.” or “Leishmania spp.”
3) Toxoplasmosis (line 63): “promising model” or should we already consider them as “efficient experimental model”? Considering the amount of studies published with T. gondii in the last 20 years, maybe the second adjective is more appropriate.
4) Line 96: Put C. callosus in italics
5) Line 223: “…are some of the main reasons…”
6) Line 230: Is not true that “most descriptions of T. cruzi infection in rodents occur in Rattus rattus”. The use of mice is much more common. A search for terms on PubMed results in 5 times more T. cruzi studies with Mus musculus than with Rattus rattus.
7) Lines 290/291: This bibliography is from 1979 and is not from an experimental study, but from an investigation of natural infection.
8) Correct reference 71. The name of the book is “Vetores da doença de Chagas no Brasil” and the editor was Galvão, C.
Author Response
MAJOR POINTS
Point 1: My main recommendation for authors is to include a topic of difficulties and challenges for the maintenance of a wild species in captivity. These challenges include the environmental and biosafety legal issues involved in the maintenance of a wild species in captivity, the field capture of the reproductive matrices (probably not in a single expedition), their transport, the adaptation, quarantine and maintenance in captivity, the necessary environmental enrichment, the care to avoid endogamy and the differences in the bioterism of this species compared to others conventionally used in scientific research, among others.
Response 1: This recommendation was accepted, and it can be observed in item 6 of the new version of the article.
Point 2: The definition of what constitute a reservoir host depends on many variables, such as geographical origin, interaction with other mammals, presence of vectors, distribution within forest strata, environmental characteristics, etc. So, defining a reservoir is much more than just finds it infected. And this aspect was not yet described in a definitive manner for T. cruzi infection in C. callosus. Thus, I suggest authors to exclude the word "reservoir" when describing the infection by T. cruzi, keeping only "natural host". Revise it throughout the manuscript, including lines 17/18 (abstract), 38 (introduction) and 226/227 (Chagas disease topic).
Response 2: This recommendation was accepted, and it can be observed in the new version of the article.
Point 3: Leishmaniasis: The article that cites Leishmania donovani chagasi is from 1984 and this is no longer a valid name. Considering the nomenclature described in the article and the origin of the sample (Maranhão State, Brazil) we can state that the Leishmania species used in the experimental infections was Leishmania infantum. Also Leishmania mexicana amazonensis is no longer a valid name for the species. The valid name is Leishmania amazonensis. Correct them in the manuscript (this topic and Table 1).
Response 3: This recommendation was accepted, and it can be observed in the new version of the article.
Point 4: Chagas Disease - Lines 200/201: Although T. infestans was the main vector of the disease for many decades in Brazil, the insect examined by Carlos Chagas was a Panstrongylus megistus. I suggest reviewing and citing the original article in this paragraph (Chagas, 1909). In the same sentence: “the life cycle” is of the parasite and not of the disease. Correct it.
Response 4: This recommendation was accepted, and it can be observed in the new version of the article.
Point 5: Trypanosoma (Herbesoma) marie (lines 235-240): First the correct subgenus name in the description is Herpetosoma. Second, this supposed new species was described at a time when descriptions were mostly based on morphological characters of the parasites, without confirmatory biochemical or molecular tests. This particular case is probably an erroneous description of a species that is not valid, given that this is the only description and the only study to date (more than 40 years later) that cites this Trypanosoma species. Moreover, the characteristic of an abrupt drop in parasitemia is commonly described for all species within Herpetosoma subgenus and the species found by the authors is probably Trypanosoma lewisi. This cosmopolitan parasite was considered species-specific for a long time and for this reason, a new nomenclature was sought for each new host. Currently, it is known that T. lewisi is capable of infecting many other hosts besides Rattus rattus, including wild rodents and primates, including man. Therefore, the course of infection does not point to a parasite-host specificity (line 240). This critical analysis has to be presented by the authors in this review.
Response 5: This recommendation was accepted, and it can be observed in the new version of the article.
Point 6: Table 1 provides a methodological detail that is not important for the review. I suggest present this table as a supplementary material. In addition, it is important to correct the taxonomic status of the parasites as described above and to revise scientific names in italics.
Response 6: This recommendation was accepted, and it can be observed in the new version of the article.
MINOR POINTS
Point 1: Keywords: Put scientific names in italics. Trypanosoma, not Tripanosoma
Response 1: This recommendation was accepted, and it can be observed in the new version of the article.
Point 2: Figure 1: Replace “Leishmania ssp” by “Leishmania sp.” or “Leishmania spp.”
Response 2: This recommendation was accepted, and it can be observed in the new version of the article.
Point 3: Toxoplasmosis (line 63): “promising model” or should we already consider them as “efficient experimental model”? Considering the amount of studies published with T. gondii in the last 20 years, maybe the second adjective is more appropriate.
Response 3: This recommendation was accepted, and it can be observed in the new version of the article.
Point 4: Line 96: Put C. callosus in italics
Response 4: This recommendation was accepted, and it can be observed in the new version of the article.
Point 5: Line 223: “…are some of the main reasons…”
Response 5: This recommendation was accepted, and it can be observed in the new version of the article.
Point 6: Line 230: Is not true that “most descriptions of T. cruzi infection in rodents occur in Rattus rattus”. The use of mice is much more common. A search for terms on PubMed results in 5 times more T. cruzi studies with Mus musculus than with Rattus rattus.
Response 6: This recommendation was accepted, and it can be observed in the new version of the article.
Point 7: Lines 290/291: This bibliography is from 1979 and is not from an experimental study, but from an investigation of natural infection.
Response 7: This recommendation was accepted, and it can be observed in the new version of the article.
Point 8: Correct reference 71. The name of the book is “Vetores da doença de Chagas no Brasil” and the editor was Galvão, C.
Response 8: This recommendation was accepted, and it can be observed in the new version of the article.
Reviewer 2 Report
The purpose of this review is to describe Calomys callosus as an interesting experimental animal model applied to parasitic diseases investigation. Although the topic of this review is of main of interest, as presented, for the manuscript indeed, I have major concerns as followed:
- Why do the authors focus on these four parasitic diseases, including 1 Apicomplexa, 2 trypanosomatids and 1 helminth, as much as the authors only talk about toxoplasma gondii in the introduction section? Physiopathologically speaking these 4 parasites are very different. The authors choice must be clearly explained.
- The authors present this in vivo model but they forget the already existing models. Indeed, what is the main model usually used to study these parasitosis? for each of the parasites, the already existing models (in vitro and in vivo) are not presented in the manuscript, as well as the advantages and disadvantages of the latter in comparison with calomys callosus. Why calomys callosus is an interesting alternative to existing models? Moreover, regarding calomys callosus studies in Table 1, many protocols seems to exist. Is there a better one? on what criteria? How to compare studies with such diverse protocols? Do the authors have experience with calomys callosus model? on the 4 parasitosis? And can the authors propose a standardized protocol in the context of this review?
- Regarding the PRISMA guidelines, informations are missing: The Title do not identify the report as a systematic review; The methods section (elegibility criteria, sources, strategy, selection process…) and the flow diagram are absent.
- The figure 1 is not appropriate to the introduction section and it needs to be improved graphically. Indeed, as it is presented, this figure is neither more nor less than a table without reference. Do the authors have the rights/authorization of use to the images used? This figure is important and needs to be improved.
- Three quarters of the cited references are more than 10 years old (73/99). Can the authors justify it? Are there any more recent scientific publications on the subject?
Author Response
Point 1: Why do the authors focus on these four parasitic diseases, including 1 Apicomplexa, 2 trypanosomatids and 1 helminth, as much as the authors only talk about toxoplasma gondii in the introduction section? Physiopathologically speaking these 4 parasites are very different. The authors choice must be clearly explained.
Response 1: Our review has as main objective to present the applicability of the experimental model Calomys callosus in the study of different parasitic diseases of public health importance. Diseases that burden health systems and cause significant mortality. Therefore, aiming to illustrate this variability in the application of the model, we used four parasites with marked pathophysiological differences, being one Apicomplexa, two Trypanosomatids and one Helminth).
In the introduction (item 1), we can see a brief account of the experimental model to be discussed in the paper and cite the parasites that will also be addressed. Next, an item is presented to discuss each parasite. In this sense, we understand that we present all the parasites in the article, and not only T.gondii.
Point 2: The authors present this in vivo model but they forget the already existing models. Indeed, what is the main model usually used to study these parasitosis? for each of the parasites, the already existing models (in vitro and in vivo) are not presented in the manuscript, as well as the advantages and disadvantages of the latter in comparison with calomys callosus. Why calomys callosus is an interesting alternative to existing models? Moreover, regarding calomys callosus studies in Table 1, many protocols seems to exist. Is there a better one? on what criteria? How to compare studies with such diverse protocols? Do the authors have experience with calomys callosus model? on the 4 parasitosis? And can the authors propose a standardized protocol in the context of this review?
Response 2: The objective of this review is to show the species Calomys callosus as an interesting experimental model for the study of the parasitic diseases presented in the paper. We fully understand the importance of other models such as in vitro and other species. Still, we believe that it would be important to present other models used if our intention was to present different experimental models for the study of parasitosis. However, our objective is to present the species Calomys callosus.
We do not understand that Calomys callosus is an alternative to the other species, but an additional option, with advantages and disadvantages. Since any experimental model represents the materialization of part of reality, it is up to the researcher to have the knowledge to choose the model that best answers that specific experimental question. We understand that there is no ideal experimental model that will answer all questions related to the study of a given disease.
In table 1, we aim to illustrate the most frequent responses found in the articles that used the model for each of the diseases, but not to compare, because it was not defining criteria that would provide us with the ability to compare diseases caused by such different parasites, in terms of pathophysiology.
The authors' experience with the model can be evidenced in some publications, but above all when observing that most authors work directly with the creation and experimentation of C. callosus. The team at Rodents Animal Facilities Complex works together with immunologist, parasitologist and cell biology researchers, knowing in depth the model in different diseases. Others have publications in renowned journals using the species. Still, it is important to emphasize the intense review of works carried out for this review.
Point 3: Regarding the PRISMA guidelines, informations are missing: The Title do not identify the report as a systematic review; The methods section (elegibility criteria, sources, strategy, selection process…) and the flow diagram are absent.
Response 3: This review is not a systematic review. Therefore, PRISMA guidelines were not presented.
Point 4: The figure 1 is not appropriate to the introduction section and it needs to be improved graphically. Indeed, as it is presented, this figure is neither more nor less than a table without reference. Do the authors have the rights/authorization of use to the images used? This figure is important and needs to be improved.
Response 4: We understand that figure 1 fulfills its objective, to briefly illustrate the most common phenotypes found in experiments on toxoplasmosis leishmaniasis, Chagas disease and schistosomiasis using C. callous as an experimental model. It was constructed by an image (photo with elements from Mind the graph software, and a table with the main phenotypes found in the reviewed papers. The idea is simply to illustrate item 1 of the article, giving the reader notes of what he will find in the other topics, where these phenotypes can be found and properly referenced.
The images used are from the authors themselves and were obtained from animals undergoing experimentation with approval by the ethics committee on the use of animals (Protocol number A001/22).
Point 5: Three quarters of the cited references are more than 10 years old (73/99). Can the authors justify it? Are there any more recent scientific publications on the subject?
Response 5: If we do a simple search in a database such as PubMed, from 1965 to 2022, using the word: Calomys callosus, we will observe that the largest amount of works with this species are between the years 2001 and 2009. Given the experience of our group with experimental models, we attribute this reduction to the small number of animal facilities that currently maintain this species. This reduced number of installations is due to different reasons, including legal ones, which strongly regulate the collection of wild animals for use in experiments. In this sense, our idea is to show in this article important aspects about the use of C. callosus in experimental protocols with important parasitic diseases, as well as to encourage that facilities that have this species already adapted in captivity can be suppliers for researchers who demand, favoring scientific development in a collaborative way and preserving the fauna.
Reviewer 3 Report
The review is interesting and addresses a critical issue, an animal model for parasitic and infectious diseases. However, it seems more like a list of studies and in my point of view, the review lacks a critical discussion of the studies described. For example, it isn´t clear why this model is underused. The authors should explore more this issue and answer several questions such as “Is it possible to implement this model outside of rodent native areas?; Why is this model underused? Because is it difficult to find the rodent?; Is the rodent susceptible to other forms of Schistosoma?”
Another point of view that should be cleared is why this model should be chosen instead of the other that already exist (eg. CD1, Balb/c, etc). I think the authors could improve the review and render it more interesting and appealing.
Line 36: Please rephrase this sentence. It is too long.
Line 246: Full stop should be after reference number.
Line 275: Please consider rephrasing the sentence. For example, “After larval skin penetration, the larvae develop into adult schistosomes which live in the blood….”.
Line 279: Include bladder cancer. Urogenital schistosomiasis is linked to the development of bladder cancer.
Line 292: Please rephrase. It is not written correctly.
Line 295: “cercariae from human isolate”. What do you mean by this? It was interesting if the authors discussed the studies and present a point of view.
Line 309: route instead rout.
Table 1: Please correct the table formatting.
Author Response
Point 1: The review is interesting and addresses a critical issue, an animal model for parasitic and infectious diseases. However, it seems more like a list of studies and in my point of view, the review lacks a critical discussion of the studies described. For example, it isn´t clear why this model is underused. The authors should explore more this issue and answer several questions such as “Is it possible to implement this model outside of rodent native areas?; Why is this model underused? Because is it difficult to find the rodent?; Is the rodent susceptible to other forms of Schistosoma?”
Response 1: We also believe in this experimental model importance, which motivated us to produce this paper. Our goal in this article was to present studies carried out with different parasite infections, different phenotypes observed with the model and mainly encourage C. callosus use. A further discussion would imply in detailing all works cited throughout the review and a comparison with other experimental models, including among the parasites addressed in the text, out of the focus of the article.
Difficulties in using C. callosus as an experimental model are mostly linked to its capture, maintenance and breeding, since they are species of wildlife. These difficulties have been approached in this new version in the topic 6 “Challenges in maintaining C. callosus as an experimental model”. We would also like to present that some animal facilities in Brazil, including our group, maintain and breed this species for many years, under sanitary and genetic control. We even have unpublished data on the genetic sequencing of our “lineage”, which will be published soon.
Point 2: Another point of view that should be cleared is why this model should be chosen instead of the other that already exist (eg. CD1, Balb/c, etc). I think the authors could improve the review and render it more interesting and appealing.
Response 2: As presented earlier, the choice of an experimental model must consider many factors. Each model reproduces part of the reality to be answered. It is often necessary to use different models to understand an infectious process. In this context, encouraging the use of C. callosus to the detriment of several other murine models and in different infectious processes is unfeasible. We believe that drawing attention to the model with highlights as we present it is the first step towards publicizing this important model.
Point 3: Please rephrase this sentence. It is too long.
Response 3: This suggestion was accepted and already modified in the final version of the article.
Point 4: Full stop should be after reference number.
Response 4: This suggestion was accepted and already modified in the final version of the paper.
Point 5: Please consider rephrasing the sentence. For example, “After larval skin penetration, the larvae develop into adult schistosomes which live in the blood….”.
Response 5: This suggestion was accepted and already modified in the final version of the paper.
Point 6: Include bladder cancer. Urogenital schistosomiasis is linked to the development of bladder cancer.
Response 6: This suggestion was accepted and already modified in the final version of the paper.
Point 7: Please rephrase. It is not written correctly.
Response 7: This suggestion was accepted and already modified in the final version of the paper.
Point 8: “cercariae from human isolate”. What do you mean by this? It was interesting if the authors discussed the studies and present a point of view.
Response 8: This suggestion was accepted and already modified in the final version of the paper.
Point 9: route instead rout.
Response 9: This suggestion was accepted and already modified in the final version of the paper.
Point 10: Please correct the table formatting.
Response 10: This suggestion was accepted and already modified in the final version of the paper.
Round 2
Reviewer 1 Report
Authors satisfactorily answered my questions and modified the manuscript following reviewer’s request, except for "point 6" of the minor revision. Authors maintained Rattus rattus, instead of Mus musculus
Author Response
Point 1: Authors satisfactorily answered my questions and modified the manuscript following reviewer's request, except for "point 6" of the minor revision. Authors maintained Rattus rattus, instead of Mus musculus.
Response 1: We thank the reviewer for the important contributions. We apologize in relation to item 6. I inform you that the exchange of Rattus rattus for Mus musculus was carried out.
Reviewer 2 Report
Point 1: Why do the authors focus on these four parasitic diseases, including 1 Apicomplexa, 2 trypanosomatids and 1 helminth, as much as the authors only talk about toxoplasma gondii in the introduction section? Physiopathologically speaking these 4 parasites are very different. The authors choice must be clearly explained.
Response 1: Our review has as main objective to present the applicability of the experimental model Calomys callosus in the study of different parasitic diseases of public health importance. Diseases that burden health systems and cause significant mortality. Therefore, aiming to illustrate this variability in the application of the model, we used four parasites with marked pathophysiological differences, being one Apicomplexa, two Trypanosomatids and one Helminth).
Rev 1 : Ok. Please add these explenation in the introduction section.
In the introduction (item 1), we can see a brief account of the experimental model to be discussed in the paper and cite the parasites that will also be addressed. Next, an item is presented to discuss each parasite. In this sense, we understand that we present all the parasites in the article, and not only T.gondii.
Point 2: The authors present this in vivo model but they forget the already existing models. Indeed, what is the main model usually used to study these parasitosis? for each of the parasites, the already existing models (in vitro and in vivo) are not presented in the manuscript, as well as the advantages and disadvantages of the latter in comparison with calomys callosus. Why calomys callosus is an interesting alternative to existing models? Moreover, regarding calomys callosus studies in Table 1, many protocols seems to exist. Is there a better one? on what criteria? How to compare studies with such diverse protocols? Do the authors have experience with calomys callosus model? on the 4 parasitosis? And can the authors propose a standardized protocol in the context of this review?
Response 2: The objective of this review is to show the species Calomys callosus as an interesting experimental model for the study of the parasitic diseases presented in the paper. We fully understand the importance of other models such as in vitro and other species. Still, we believe that it would be important to present other models used if our intention was to present different experimental models for the study of parasitosis. However, our objective is to present the species Calomys callosus.
We do not understand that Calomys callosus is an alternative to the other species, but an additional option, with advantages and disadvantages. Since any experimental model represents the materialization of part of reality, it is up to the researcher to have the knowledge to choose the model that best answers that specific experimental question. We understand that there is no ideal experimental model that will answer all questions related to the study of a given disease.
Rev 2 : I understand the position of the authors. However, in order to discuss the advantage and disadvantage of this experimental model, the comparaison with others is of importance.
In table 1, we aim to illustrate the most frequent responses found in the articles that used the model for each of the diseases, but not to compare, because it was not defining criteria that would provide us with the ability to compare diseases caused by such different parasites, in terms of pathophysiology.
The authors' experience with the model can be evidenced in some publications, but above all when observing that most authors work directly with the creation and experimentation of C. callosus. The team at Rodents Animal Facilities Complex works together with immunologist, parasitologist and cell biology researchers, knowing in depth the model in different diseases. Others have publications in renowned journals using the species. Still, it is important to emphasize the intense review of works carried out for this review.
Rev 2 : Ok.
Point 3: Regarding the PRISMA guidelines, informations are missing: The Title do not identify the report as a systematic review; The methods section (elegibility criteria, sources, strategy, selection process…) and the flow diagram are absent.
Response 3: This review is not a systematic review. Therefore, PRISMA guidelines were not presented.
Rev 3 : Ok.
Point 4: The figure 1 is not appropriate to the introduction section and it needs to be improved graphically. Indeed, as it is presented, this figure is neither more nor less than a table without reference. Do the authors have the rights/authorization of use to the images used? This figure is important and needs to be improved.
Response 4: We understand that figure 1 fulfills its objective, to briefly illustrate the most common phenotypes found in experiments on toxoplasmosis leishmaniasis, Chagas disease and schistosomiasis using C. callous as an experimental model. It was constructed by an image (photo with elements from Mind the graph software, and a table with the main phenotypes found in the reviewed papers. The idea is simply to illustrate item 1 of the article, giving the reader notes of what he will find in the other topics, where these phenotypes can be found and properly referenced.
The images used are from the authors themselves and were obtained from animals undergoing experimentation with approval by the ethics committee on the use of animals (Protocol number A001/22).
Rev 4 : Ok.
Point 5: Three quarters of the cited references are more than 10 years old (73/99). Can the authors justify it? Are there any more recent scientific publications on the subject?
Response 5: If we do a simple search in a database such as PubMed, from 1965 to 2022, using the word: Calomys callosus, we will observe that the largest amount of works with this species are between the years 2001 and 2009. Given the experience of our group with experimental models, we attribute this reduction to the small number of animal facilities that currently maintain this species. This reduced number of installations is due to different reasons, including legal ones, which strongly regulate the collection of wild animals for use in experiments. In this sense, our idea is to show in this article important aspects about the use of C. callosus in experimental protocols with important parasitic diseases, as well as to encourage that facilities that have this species already adapted in captivity can be suppliers for researchers who demand, favoring scientific development in a collaborative way and preserving the fauna.
Rev 5 : Ok.
Author Response
Point 1: Why do the authors focus on these four parasitic diseases, including 1 Apicomplexa, 2 trypanosomatids and 1 helminth, as much as the authors only talk about toxoplasma gondii in the introduction section? Physiopathologically speaking these 4 parasites are very different. The authors choice must be clearly explained.
Response 1: Our review has as main objective to present the applicability of the experimental model Calomys callosus in the study of different parasitic diseases of public health importance. Diseases that burden health systems and cause significant mortality. Therefore, aiming to illustrate this variability in the application of the model, we used four parasites with marked pathophysiological differences, being one Apicomplexa, two Trypanosomatids and one Helminth).
Rev 1 : Ok. Please add these explanation in the introduction section
Response 1: Suggestion accepted. This explanation was inserted in the introduction. We appreciate and understand that it contributed to the improvement this review.
Point 2: The authors present this in vivo model but they forget the already existing models. Indeed, what is the main model usually used to study these parasitosis? for each of the parasites, the already existing models (in vitro and in vivo) are not presented in the manuscript, as well as the advantages and disadvantages of the latter in comparison with calomys callosus. Why calomys callosus is an interesting alternative to existing models? Moreover, regarding calomys callosus studies in Table 1, many protocols seems to exist. Is there a better one? on what criteria? How to compare studies with such diverse protocols? Do the authors have experience with calomys callosus model? on the 4 parasitosis? And can the authors propose a standardized protocol in the context of this review?
Response 2: The objective of this review is to show the species Calomys callosus as an interesting experimental model for the study of the parasitic diseases presented in the paper. We fully understand the importance of other models such as in vitro and other species. Still, we believe that it would be important to present other models used if our intention was to present different experimental models for the study of parasitosis. However, our objective is to present the species Calomys callosus.
We do not understand that Calomys callosus is an alternative to the other species, but an additional option, with advantages and disadvantages. Since any experimental model represents the materialization of part of reality, it is up to the researcher to have the knowledge to choose the model that best answers that specific experimental question. We understand that there is no ideal experimental model that will answer all questions related to the study of a given disease.
Rev 2 : I understand the position of the authors. However, in order to discuss the advantage and disadvantage of this experimental model, the comparaison with others is of importance.
Response 2:
We appreciate this timely reviewer's suggestion. We agree that there are other experimental models to study the mentioned parasitic diseases, such as models involving the use of cell lineage, as well as other animal species. However, we understand that the current literature has already many works on these other experimental models (as mentioned below). For this reason, we propose, for the first time, to compile a robust review article addressing a current view of the remarkable findings involving the use of Calomys callosus as an interesting experimental model to investigate parasitic illness.
Thus, we strong believe that this peculiar hot topic deserves a robust review, which can be addressed on the Special Issue “Advances in Parasitic Diseases” in the Section “Parasitic Pathogens”, highlighting the most recent advances in this field.
Reviewer 3 Report
The authors addressed all the questions.
Author Response
Thanks for all the notes. They greatly enriched our work.